# Impact of Corporate Political Activity on the Relationship Between Corporate Social Responsibility and Financial Performance: A Dynamic Panel Data Approach

**Woon Leong Lin** [1] , **Jo Ann Ho** [1,*] **and Murali Sambasivan** [2]

1   Faculty of Economics and Management, Universiti Putra Malaysia, 43400 Selangor, Malaysia; linwoonleong@gmail.com
2   Lakeside Campus, Taylor's University, 47500 Selangor, Malaysia; sambasivan@hotmail.com
*   Correspondence: ann_hj@upm.edu.my

**Abstract:** As corporate social responsibility (CSR) gains momentum in the business world, it is imperative to comprehend the relationship between CSR and corporate financial performance (CFP). While there is prior research looking at this relationship, scholars have proposed a contingency view that is meant to determine the situational contexts in which critical associations between CFP and CSR activities will arise. This study provides further insight into the moderating effects of corporate political activity, specifying the ways in which different arrangements of corporate CSR and CPA might align or otherwise, thus influencing CFP beyond associated dissimilar effects on corporate performance. The data for this study was obtained for the periods 2007–2016 from the samples selected from the list of Fortune's World's Most Admired Companies. The dynamic panel data was analyzed using the System Generalized Method of Moment estimation. The main findings are that CSR does not significantly influence CFP. However, CPA does negatively moderate the relationship between CSR and CFP. This indicates that high political expenditures worsen a firm's financial position compared to the financial position of firms with less spending on CPA.

**Keywords:** corporate political activity; corporate social responsibility; corporate financial performance; stakeholder theory; system GMM

## 1. Introduction

The practice of corporate social responsibility (CSR) has increasingly gained popularity among organizational managers, since firms are being assessed based on their societal contributions as well as financial performance. Many corporations undertake socially responsible initiatives in order to derive legitimacy beyond mere competitive advantage, although research on the association between CSR and firm performance has produced mixed findings, with potential relationships remaining indefinable and doubtful [1]. Much research has involved inconsistent outcomes, owing mainly to a disregard for contingency variables as well as errors in measurements, wrongly specified models, undersized sampling across multiple industries, singular CSR dimensions, and an indefinably broad operationalization of CSR and corporate financial performance (CFP) aspects [1–3]. Such factors considerably affect the connections between these activities. Nonetheless, investigators have started to doubt the continuousness of studies on CSR [4]. One material point remains in question—what can make corporate social responsibility actions significantly affect firm financial performance? To address

this issue, this study covers the emerging significance of the moderating role of corporate political activity (CPA) as a core area of study.

Over many decades, increasing concerns have emerged regarding political factors in CSR [5,6]. Corporate philanthropy has translated into further CPA activity, particularly among European firms, as organizations increasingly assume responsibilities that were the traditional domain of governments [7]. This trend partly involves firms that engage in philanthropic CPA, which is often considered a form of lobbying, in order to effect more beneficial corporate or societal conditions. Thus, one can see an increasing interest in political corporate social responsibility, or CSR–CPA interaction, which is defined as an activity where CSR programs feature unintended or intended political effects [8]. The term "corporate political activity" refers to a firm's attempts to shape government policy in ways favorable to the firm [9]. Reviews of both bodies of literature are appropriate as well as essential. Such studies have outlined many CSR influences on CPA, as exemplified in the socio–political role of firms in providing community services, which can include health and education programs, which were traditionally the responsibility of the governments [10,11]. Similarly, the emerging role of self-regulating corporate behaviors via voluntary initiatives, in terms of globalized codes of conduct and eco-friendly volunteer programs, fills gaps in globalized governance and national regulations [12,13]. Empirical investigations propose that corporations consistently apply CPA by moving toward changes in regulations, relative to societal and environmental problems, via lobbying efforts, participation in advisory groups, and other conventional socio–political channels [14].

Up to this point, little research has examined how indirect factors, such as CPA, affect the relationship between CSR and CFP in the business environment. Hence, the main contribution of this study is to extend CSR literature by laying out the moderating effect of CPA's influence on the CSR–CFP nexus and analyzes how this moderation operates through synergies and trade-offs, which previous studies have disregarded.

By utilizing panel data of the 100 Fortune's World Most Admired Companies (WMAC) 2007–2016, this paper focuses chiefly on the relationship between CFP and CSR, in an effort to answer the following questions. First, what role does CSR play in contributing to a firm's performance in WMAC? Second, does CPA contribute to the weakening or strengthening of the relationship between CFP and CSR? In this research, existing models are employed for extrapolation, following Waddock and Graves [15] and McWilliams and Siegel [16]. The type of relationship between CSR efforts and the performance of the company is explored. The results would provide firm-level evidence defining the relationship between CSR and the performance of the company and determining whether that complements the recent theoretical literature regarding microeconomics, pertaining to the performance model of CSR-based endogenous firms.

In this study, the System Generalized Method of Moments (SYS-GMM) technique has been used for estimation, considering the endogenous CSR investments of firms. Based on the empirical results, CSR was found to cast a negative effect on CFP for WMAC under unobserved firm heterogeneity under a controlled condition. Return on Assets (ROA) is our key performance metric, which can be defined as the accounting measure of the firm, divided with earnings before tax for the assets. First, we evaluated if firms that had a high CPA, as provided by PAC (Political Action Committee) as well as lobbying expenditures, could improve the value of a firm by increasing CSR. For firms with a high CPA, a negative effect was cast by CSR activities on performance, which indicates that costs may outweigh the benefits for such firms. Conversely, for firms that had low CPA, a positive impact was seen. However, it was shown that the CPA's positive impact would be reversed for firms that had a poor prior CSR. This was measured with the help of Fortune's rating regarding the list of 'World's Most Admired Companies.' Interestingly, no evidence showing a direct link between CSR and the firm's value was found, when the firm characteristics for an unobservable time-invariant were kept in control (i.e., for models using SYS-GMM).

This study is structured in the following manner: the "Theoretical framework" portion addresses the hypothetical underpinnings and presents the key administrative models and concepts that this

research relies on. CSR, CPA, and CFP are then presented as the three main analytical elements of our conceptual framework. The "Proposed framework and formulations of hypotheses" section attempts to validate the dual basic hypotheses that may yet resolve the question of how corporate performance can be explained by CSR. In the "Empirical study" section, our theory was empirically tested using Fortune's Corporate Social Responsibility Index, which measures CSR along with the trio of financial ratios that represent corporate performance. The last section comprises the conclusion and recommends future modes of research.

## 2. Literature Review and Theory Development

### 2.1. Corporate Social Responsibility (CSR)

According to Carroll [17], definitional and conceptual constructs of modern corporate social responsibility practices originated during the 1950s. Holme and Watts [18] defined corporate social responsibility as a "corporate commitment to advance sustainable socio–economic development, while working with staff, family, the local community, and other social elements to enhance quality of life" (p. 2). McWilliams and Siegel [16] have defined such an activity as one that appears to cause progress in certain societal goods beyond the commercial interests and legal responsibilities of a corporation. Given these theoretical viewpoints, contradictory perspectives seem to be present.

In the neo-classical economic models established by Friedman [19], corporate managers who were chosen to represent stockholders are solely obligated to serve the interests of their principal stakeholders, their shareholders. It was suggested that corporations only have a social responsibility to use resources while engaging in activities that raise profits. From Friedman's perspective, all other behaviors, including charities, eco-friendly initiatives, or employee benefit enhancements, which might affect an organizations' optimal usage of resources towards the maximization of shareholder value, oppose the firm's societal responsibilities. Investigations of the negative effect of CSR on CFP [20,21] have supported this viewpoint.

Freeman [22] conversely proposed a stakeholder type of model, where organizational decisions are meant, not only to consider shareholder interests, but also those of other stakeholders, such as staff, clients, vendors, and local communities, who are thereby affected. The model assumes that CSR actions involve the claims of all legitimate stakeholders, with the purpose of improving organizational profits as well as reputation. The scheme draws support from certain empirical studies that have confirmed affirmative associations between corporate social responsibility and financial performance [23–25]. Using a progressive stakeholder model, Jensen [26] attempted to account for potential conflicts among stakeholders and the neo-classical economic models. According to this more sophisticated scheme, CSR is said to represent key corporate strategies and core competencies that can affect the value of corporations and do not only represent the prevailing altruistic behaviors or ethical needs.

### 2.2. CSR–CFP

Earlier studies concentrated on the impact of firms' CSR activities, showing that CSR could cast an impact on the reputation of a firm [27,28], brand performance [29], shareholder value [30,31], corporate innovation [32], customer satisfaction [33], minimize firm risk [34], and improve financial performance [35–37]. CSR is considered a crucial approach to producing wealth and enhancing a firm's performance [9,38]. Numerous scholars have specified that a close link exists between CSR and the company's performance [1,39], but it is still debatable whether CSR efforts could impact financial performance.

A few researchers have shown that the relationship between CSR and financial performance is positive [40–42]. Based on this perspective, Waddock and Graves [15] mentioned that CSR practices provide greater benefits when compared with the costs involved. Shin and Thai [43] suggested that a firm's social commitment results in an improvement of its economic performance. On the flipside of the equation, other researchers, particularly neoclassical economists, argue that CSR casts a negative effect

on financial performance, as they perceive that such social expenditure could have been avoided [44,45]. On the other hand, a few other scholars argue that CSR does not cast any effect or has a neutral impact on financial performance [16,46].

By investigating the effect of CSR on performance and further emphasizing more inconclusive CFP–CSR associations, such that the influence of CSR on performance may be better recognized, current studies draw attention to the critical moderating aspects of CPA. Richter [47] and Werner [48] published direct evidence in support of the associations among CSR and CPA. These activities can elevate corporate values [47] and may be mutually considered as economically complementary in their effects. Rowley and Berman [49] and Wang and Choi [50] suggested that there is a significant basis for the inclusion of moderating factors when conceiving more nuanced descriptions of the proposed CFP–CSR nexus. Many studies have demonstrated that the moderating aspects play a critical role in explicit evaluations of theoretical associations [51–53].

### 2.3. Corporate Political Action (CPA)

Socio–political roles in corporate strategies have gathered increasing relevance in recent decades [54]. Researchers have further evaluated socio–political risks, corrupt practices, and social and political connections as factors. Such links are instrumental to the performance of globalized organizations [21,55,56]. However, proximity to social and political power does entail costs, namely, in the exposure to uncertain prospects that stem from the effects of future risks and policies [57]. Key advances in the recognition of CPA were lately seen in the fields of strategic management, political, sociological, and socioeconomic sciences, and financial practice. The aspirations of corporations to engage every socio–political process spring from the realization that practically all firms are affected by public policy and regulations [58]. Other than for self-interest, corporations will also engage these activities as corporate citizens [4,59]. Corporate citizenship describes the political role that corporations play in the promotion of social welfare [60]. In fact, the political efforts made by the organizations also potentially enhance the economic sustainability of CSR [14]. Additionally, significant knowledge on how to address the existing social and environmental issues through these political connections effectively assist the CSR investments made by the organizations [14,18,61]. As reaffirmed by Peterson and Pfitzer [61], the solid connections with these governmental agencies are more likely to increase the effectiveness of addressing societal issues successfully in the CSR commitment of these organizations. With political support, the organizations may obtain and expand the resources for their CSR initiatives as well. Thus, the knowledge and endorsement from these political influences provide enhanced credibility and the necessary opportunities for the organizations to prioritize the development as well as the economic sustainability of CSR [62].

Organizations tend to follow trending globalization movements, principally multinational enterprises (MNEs) that are working to take on responsibilities that were traditionally considered to be within the sphere of government [5]. From endorsements and support for political candidates [60] to advocacy on behalf of mutually beneficial regulatory and legal schemes [63], CPA's contribution to local governance is prevalent. CPA represents a typical non-market approach to grassroots politics and lobbying, corporate donations for PACs, as well as the development of political connections with governing officials [64]. Such a "political market" framework of CPA sees businesses as "demanders", with government officials as "suppliers" of social and political favors and actions [65,66]. Demander entities provide information, funds, and assets to favored official suppliers, who in turn demand expertise on how to shape governmental policy and distribute proceeds for party elections and primary purposes. This exchange relationship features types of mutual dependencies and also relational associations, which enable businesses to achieve some degree of access to political actors, who see to it that their perspectives are recognized and favorably assessed [67]. These connections can then shape governmental policy-making that favors certain corporate objectives [68,69]. Such contacts can similarly affect official decisions in an indirect manner, in accordance with structural dependencies that typically exist within bureaucratic and legislative organizations.

## 3. Hypotheses Development

The study set out to examine the influence of CSR on CFP with CPA as the moderator. Based on the research framework (Refer to Figure 1), the following hypotheses were formulated.

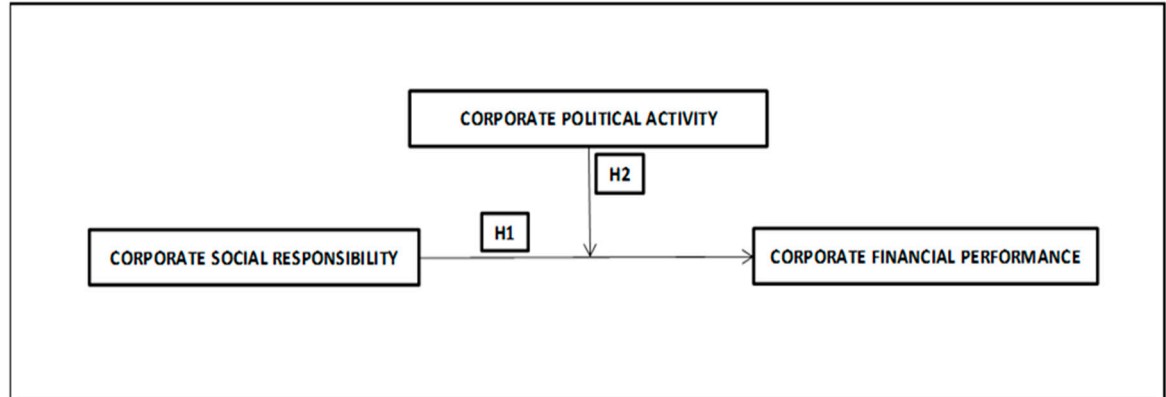

**Figure 1.** Research framework.

### 3.1. Impact of CSR on CFP: From the Perspective of Stakeholder Theory and Resource Dependency Theory

Seen from a theoretical standpoint, our research framework is based on stakeholder theory formulations. Stakeholder theory [36] first developed a framework for the relationship between CFP and CSR, stating that company resources are claimed by interest groups, which in turn results in an implicit requirement for proper company behavior, like care for impartial labor relationships and consideration for the environment. When the company fails to behave with social responsibility, the resultant costs could tend to be significant and stand for a financial burden that will most likely lessen profits, resulting in an entity that is less socially aware. On the other hand, if firms actively implement socially responsible initiatives, this will result in additional profit, and subsequently, businesses will be provided with an incentive to make socially responsible investments, thus raising investments in CSR activities [70].

Customers, suppliers, shareholders, employees, and society in its entirety stand for interest groups for the corporations. Nevertheless, it is posited by the stakeholder theory that investors tend to favor companies that possess superior social behavior, given that all the other factors are kept constant, and there is independently available information on social responsibility. It was noted that an organization's primary goal is to meet the conflicting demands issued by various stakeholders. It is further postulated by the theory that the level of effort that companies allot to the different facets of CSR varies based on the importance that each of the interest groups has to them. The stakeholder theory serves as a vital hypothetical basis for studies on CSR. It ventures beyond the conventional belief of maximizing shareholder equity, so that a new perspective for analyzing CSR can be developed [71]. We focus on the stakeholder theory to conduct an analysis of the effect that CSR has on firm performance.

Corporations are gaining greater awareness of the reputational risks and opportunities associated with CSR. This is because CSR performance is a vital factor in the stakeholders' decision to contribute resources to the firms or not [37]. Accordingly, firms that inline their business activity with their stakeholders' expectations prioritize CSR [72]. It has been suggested by Vaiman et al. [73] that a firm's CSR activities have increased in importance. According to Grimmer and Bingham [74], customers have a greater willingness to purchase services and products from firms that possess better CSR engagement. It has been emphasized by Mohr and Webb [75] that customers are willing to pay a premium price for these firms' services and products. According to Becker-Olsen et al. [76], customers are trustworthy and better impart word-of-mouth when a certain firm is believed to have better CSR performance. Investors tend to provide investment prospects to firms that are seen as decent corporate citizens [77] as they are perceived to be more sustainable and stable in development. Likewise, governments

provide more policy support and subsidies to good corporate citizens. Furthermore, it is more likely that creditors (especially banks) will lease firms that possess a greater degree of financial stability [78]. On the other hand, as suggested by Lyon and Maxwell [79], a firm that neglects CSR will likely turn out to be the target of environmentalists and regulators, which would harm its reputation and the value of its brand.

CSR enhances corporate reputation and stakeholder support, which have direct contributions to corporate value and the wealth of shareholders [80]. In earlier studies, it has been proposed that CSR promotes a firm's socially responsible public image, which tends to build favorable relationships with various stakeholders. Subsequently, this facilitates increased stakeholder support in the case of increased worker loyalty [81], substantial customer support [82,83], an increased level of legitimacy among communities [84], and an enhanced and even deeper government relationship [40]. Furthermore, CSR may offer firms insurance value, since affirmative moral capital taken from CSR can be helpful in moderating the risks of losses in shareholder values [85] every time firms experience adverse events [80,86]. Therefore, enterprises will achieve a good societal performance, which further generates an enhanced financial performance [87]. On the basis of the above arguments from stakeholder theory, the following relationship is hypothesized:

**Hypothesis 1 (H1).** *A positive relationship exists between the CSR activities of firms and CFP.*

### 3.2. Moderation Effect of CPA on the CSR–CFP Relationship

It is suggested by the resource dependency model that firms choose external dependencies to lessen uncertainties [68]. It was noted by Salancik and Pfeffer [88] (pp. 189–190) that "organizations, through CSR and socio–political mechanisms, try to secure environments that suit their interests" and that "organizations might deploy CSR-political means to vary conditions in socioeconomic settings." Therefore, there is an active attempt of firms to "create" desirable conditions by aiming to mold governmental regulations, which could result in more favorable environments. CPA is still a general concept that requires corporate strategies and procedures designed "to influence government policy or processes" [89] (pp. 32–33). One can utilize various kinds of engagements to communicate policy preferences to policymakers: monetary contributions, informational provisions, and the shaping of constituencies [66]. Corporations can use the resource base generated through their socio–political behavior, such that their CSR can be provided for in three or more ways: (a) by helping firms choose CSR priorities; (b) by improving the CSR policy's sustainability; and (c) by strengthening the CSR commitment's credibility.

Firms that aim to develop their CSR commitments must select the topics or the portions of society that they want to be profiled with. If CSR is perceived to contribute to society, then decisions involving the development CSR must take societal needs into consideration [90]. At the very least, it is expected that any CSR justification given to outside audiences should emphasize topics that are considered to be pressing societal challenges. It was established by Campbell and Slack [54] that corporations take such issues into consideration in their disclosures. Given their organizational contacts, firms may be able to hear about and develop heightened sensitivities to, as well as knowledge of, challenges and issues within the society. According to Post et al. [91], this role of CPA can be referred to as "socio–political intelligence". Informational exchanges with administrators and politicians can assist firms in identifying problems that have both social and political relevance. This tends to take place whenever firms implement "relational" approaches to CPA [66], because they enable informational exchanges and allow firms to be receptive to the problems, perspectives, and concerns of their respective regulatory and socio–political counterparts. Corporate socio–political activities may also be helpful in enhancing the market viability of CSR techniques (i.e., assisting in designing and executing external action programs) [91] (p. 139). Corporate political connections could enhance the efficacy of CSR investments by providing vital knowledge for addressing environmental and social problems.

Firms that take advantage of their government connections to help with their socially responsible activities tend to successfully address social issues [61], because they are given better information on the kinds of activities that the society needs. Furthermore, connected firms obtain specific advantages from their political connections [44]. We therefore predict that CPA has an affirmative moderating effect on the relationships between CFP and CSR. Together, CSR and CPA can generate complementary corporate-level resource bases that would allow firms to maximize the effectiveness of both programs simultaneously. These kinds of perspectives involve CSR–CPA complementarity in responding to governing pressures, which one can use to defend corporate legitimacy and accelerate the influx of vital government-controlled resources. The following hypothesis can therefore be proposed:

**Hypothesis 2 (H2).** *The relationship between CSR and CFP is moderated by the firm's investments in CPA; the positive relationship between CFP and CSR tends to be stronger with a greater investment in CPA.*

## 4. Sample Data

In the initial sample, all the firms in the 2016 Fortune list of the world's most admired companies were included. In 2016, a total of 342 companies were listed. To avoid issues in sampling selection, there was no mandate requiring a balanced panel, with a minimum of five years being consecutively listed in the Fortune magazine. Thus, the sampling numbers varied annually. Consequently, the estimation approach utilized as many available observations as possible. Moreover, to develop the database's dimensional dynamics through the introduction of the dependent variable's lagged values, firms were observed for no less than two consecutive years. Firms that did not present full information were excluded. To improve the conditions underlying the assumptions of the study, Cook's distance was used to determine the outlier and influential cases. Influential cases that had distance values of more than 1 Anderson, Sweeney and Williams [92] were eliminated. The study obtained a final and unbalanced panel sampling that had 1294 observations. These observations represented 134 firms annually for the period from 2007 to 2016. For this analysis, the panel data of CFP and CSR were utilized, based on the obtained information, as surveyed annually by the Fortune magazine in the production of its popular list of the "World's Most Admired Companies". CSR was determined from the WMAC list. The financial information for these explanatory variables was obtained from Thomson Reuters Datastream. Furthermore, CPA information was obtained from mandated public disclosures of corporate expenditures used for lobbying. The basis of this is the research conducted by Richter, Timmons, and Samphantharak [93]. This work encoded and cleaned public records from the Centre for Responsive Politics data in ways that made it possible for the information to be combined with other datasets that have additional information about these firms.

## 5. Variables

### 5.1. Dependent Variable—CFP

Empirical research examining the connections among CFP and CSR has often utilized different financial performance measures. In particular, Griffin and Mahon [94] determined about 80 measures of corporate performance. These were then adapted to a set composed of 51 papers. Some of the more widely-utilized are the accounting-based profitability measures [16,95], such as return on assets (ROA), return on equity (ROE), and net profit, as well as market-based indicators, such as earnings per share, Tobin's Q, market returns, and market-to-book values [96]. In general, accounting-based measures are considered to be representative of short-term or past performance, in financial terms. Alternatively, market-based indicators stand for the future (i.e., long-run performance) [97]. According to the proponents of these financial measures, numerous factors that are unrelated to corporate activity can affect market-based indicators. However, these proponents disagree with the objectivity of accounting-based figures. As a result, they emphasize the importance of value-based measures connected to investor and shareholder interests [98]. On the other hand, Griffin and Mahon [94]

emphasized the significance of implementing traditional accounting-based measures of performance, since modern, value-based measures could be reflective of more than just solely economic performance. Regardless of the limitations of these economic measures, they have been widely accepted as measures that can offer the most accurate representation of the financial aspects of corporate performance [95]. Furthermore, the results obtained from earlier research support the argument that CSR tends to exhibit stronger associations with accounting-based returns compared to investor-related returns [25]. Given these arguments, this study therefore makes use of proxies for financial performance. Furthermore, three kinds of accounting-based performance measures are salient, including the previously mentioned ROA. ROA is a way to measure the after-tax net income level plus interest, before the preferred dividends are included, and in terms of prior-year assets and per-unit averaged current assets. Numerous researchers have employed this measure [99–102].

*5.2. Independent Variables*

5.2.1. CSR Measure and Data

For the years 2007 to 2016, CSR was determined based on the Fortune WMAC list. Prior research has used this index extensively to measure CSR [34,36,103]. The result can then be defined as the overall CSR performance of a firm, compared to its leading industry rivals. Studies in finance [1], marketing [104,105], and strategy [36] provide details of the methodology. This archival-quality source is generally accepted as valid and reliable. Houston and Johnson [104] believe that it is the best extant secondary information source. Furthermore, the WMAC list is comprehensive, in terms of its CSR measurements, since it gathers data from 10,000 directors, securities analysts, and executives. The sampling is representative of the Fortune 1000 list of large firms across 51 industries, which have been ranked based on sales. This ranking is conducted every Fall (since 1982), and the summary results are released every March. These include the largest firms in as many as 25 industrial groups, with ratings that correspond to the largest 10 in every group. For each firm-year observation, the rating of CSR is conducted through an interval, which varies on a scale of 0 to 10. In the Fortune survey, each firm is rated, relative to its respective leading competitors, in terms of nine traits. It utilizes an 11-point scale, with 0 being poor and 10 being excellent. The nine qualities include financial soundness, long-term investment value, firm-wise usage of assets, service and product quality, management quality, innovativeness, the capacity to draw, develop, and maintain talented workers, global competitiveness, and environmental and community responsibilities.

5.2.2. CPA Measure and Data

It has been determined that CPA reflects the non-market-related behaviors of firms. These behaviors include the following: aids to PACs, current socio–political relationships among the firm and policymakers, like politically-connected directors, staff, or shareholders, and hired or in-house lobbying activities and any other related programs. Included as well are the reported testimonies or petitions to or with regulatory officials. This is at the top of the political activities associated with trade, as well as umbrella associations and organizations. PAC support and lobbying cover the two main avenues for U.S.-based firms to get closer to the two chambers of Congress. These kinds of contributions, as revealed to the public through the Lobbying Disclosure Act of 1995, are unconstrained. They also aim to advance corporate perspectives of the institutional landscape where firms operate. For this part of the study, the source was the Centre for Responsive Politics, which is responsible for directly gathering data from the semi-annual reports, filed by firms with the Senate Office of Public Records. This area has been covered by the Centre since 1998. The PACs that firms and other special interests establish are explicitly meant to oppose or support candidates for elections. The treasury of a firm can make contributions to the operating expenses of PACs. However, offering additional support is not allowed. All the proceeds must come from third parties. To achieve this, firms will often resort to primary constituents like shareholders, workers, and so on.

*5.3. Control Variables*

A challenge in the evaluation of corporate performance is the need for controlling alternative explanations. Several relevant variables need to be considered at the corporate level. First, firm size is included. This is measured in total assets, which is taken from the natural logarithm of the asset base, represented in US dollars. It is also measured through the total revenue, which is obtained from the natural logarithm of the corporate annual net revenue base. According to Dang, Li, and Yang [106], the better firms control the firm's size in its product market, while using the total revenue and the total resources from which the firm can generate profit. Thus, the total assets, as a proxy, are more relevant. Second, it should also consider the organizational slack that affects CSR [107]. Third, the determination of a further metric is also achieved via the free cash ratio, which serves as another factor [108]. Fourth, it also includes leverage ratios, since it has been demonstrated by researchers that they can influence CFP [91,109]. Finally, assessments have also incorporated advertising expenses, since they have also been proven to affect CSR [91]. All the required corporate information was supplied by DataStream.

## 6. Empirical Strategy

This empirical exercise inspects the possible effect that CSR has on CFP. It also examines the interactions among CSR and CPA on CFP. A two-step method was developed to test for interactions among CSR and CPA that influence associations, in terms of CSR and financial performance (determined by ROA). During the first stage, the effect of CSR scores on performance was estimated via the system GMM method. This was performed in order to develop the dataset's dynamic dimensions. As the second stage, a test was conducted to determine the presence of complementary CPA and CSR interactions. It was also used to identify the relative substitute inputs of economic performance. This empirical method made it possible to handle a trio of challenges. First, the dynamic structure of the dataset was exploited by considering the high likelihood that the present performance and extent of CSR can be explained by past performance (see [110]). This study accounted for the fact that there is a tendency for the present performance to have a correlation with factors that are either observable or non-observable, such as heterogeneity. Similarly, these factors could establish CSR decisions. This allowed for the involvement of corporations that were especially dependent on higher-quality processes and products, as well as corporations that had a greater likelihood of committing more heavily to CSR. Thus, this study partially corrected for such endogeneity issues by not readily overstating the contributions to financial performance [111]. Thirdly, this study took into consideration the fact that causality could run in either direction. In other words, it could move from CFP to CSR or from CSR to CFP.

Earlier studies [112–114] were followed—in particular, the existing relationships between the CSR variables ($CSR_{it}$) and financial performance ($CFP_{it}$ (ROA)), including the set of firm-level control variables (total revenue, total assets, advertising ratio, slack, and leverage) or $CTRL_{it}$. This study considered the following equation:

$$CFP_{it} = \alpha + \beta_1 CFP_{it-1} + \beta_2 CSR_{it} + \beta_3 CPA_{it} + \beta_4 CSR_{it}*CPA_{it} + \beta_5 CTRL_{it} + Year_{it} + \varepsilon_{it} \quad (1)$$

where i represents the firm, t stands for the time, $\mu_i$ represents the firm-specific fixed effects that remained unobserved, and $\varepsilon_{it}$ represents the error term, which was distributed as N $(0, \sigma^2)$. $CFP_{it}$ stands for the metrics of financial performance, which is ROA. $CSR_{it}$ represents the total CSR score. On the other hand, $CPA_{it}$ represents the firm's total lobbying expenditures. This study also took into consideration several additional variables ($CTRL_{it}$) to explain the cross-sectional variations in ROA, such as the ability to explain the financial performance. The following control variables [110,115,116] were taken into consideration: total asset, total revenue, leverage, slack, and advertising. The year represents the year-fixed effects, which are responsible for controlling any common CSR trend over time. During the estimation of Equation 1, three endogeneity sources may emerge: simultaneity (when the independent variables work as a function or as the expected values of the dependent variable),



unobservable heterogeneity (when the unobservable factors are affected by both the dependent and explanatory variables), and current CSR values, which have their basis in past CFP (an often-ignored cause of endogeneity). To remove the effects of endogeneity, the system GMM or the dynamic panel Generalized Method of Moments (GMM) estimator was utilized [117]. This was supported by Li [71], who posited that GMM has the greatest correction effect on the coefficient. In addition, Li [71] commented that the dynamic GMM potentially corrects an upward bias, in OLS estimation, of a dynamic model under reasonable conditions. It also corrects a downward bias in the mean-difference estimation of a dynamic model, if time period T is small.

## 7. Results and Discussion

The experiment was conducted in two steps. First, it tested the direct evidence of the dynamic relationship observed between CSR scores and financial performance (ROA). Second, it directly tested the effect of the interaction between CSR and CPA on financial performance. Tables 1 and 2 present a summary of the descriptive statistics for each of the model's variables. Even though the pairwise correlations observed between the independent variables were not predominantly high, it was still found that the total assets are correlated with other variables, where it is possible that a multi-collinearity concern will arise. After conducting a more detailed examination of the variables' variance inflation factor (VIF), no serious multi-collinearity concerns were revealed. A mean VIF value of 1.56 and a maximum VIF value of 2.24 were observed.

**Table 1.** Descriptive statistics N = 134 cross-firms. T = 2007–2016.

| Variable | Unit of Measurement | Obs | Mean | Std. Dev | Min | Max |
|---|---|---|---|---|---|---|
| Return on Asset (ROA) | Ratio of net income before extraordinary items to total asset of firm | 1294 | 7.0010 | 5.8409 | −35.4900 | 45.2900 |
| Corporate social responsibility (CSR) | Scale from 0 to 10 | 1294 | 6.7892 | 0.7018 | 4.2100 | 8.8000 |
| ln CPA (corporate political activity) | Log of total amount of lobbying | 1294 | 7.4189 | 1.8344 | −2.3026 | 10.6349 |
| Leverage | Ratio of long-term debt of firms to their total equity | 1294 | 5.0669 | 28.5957 | 1.1200 | 991.2100 |
| Slack | The percentage of free cash flow to the amount of sales | 1294 | 9.5095 | 16.5580 | −208.3600 | 291.1100 |
| Advertising | Ratio of advertising | 1252 | 21.2838 | 14.1068 | 0.2700 | 118.4800 |
| ln Total Assets | Log of total assets of firm | 1294 | 10.4596 | 1.2666 | 7.2821 | 13.6324 |
| ln Revenue | Log of total net sales of firm | 1294 | 10.1582 | 1.2061 | 4.1109 | 13.0949 |
| Dummy Advertising | Dummy variable = 1 if firm do not have reported advertising expenses | 1294 | 0.9946 | 0.0734 | 0.0000 | 1.0000 |

Notes: All statistics are based on original data values.

**Table 2.** Correlation.

| | 1 | 2 | 3 | 4 | 5 | 6 | 7 | 8 |
|---|---|---|---|---|---|---|---|---|
| ROA | 1 | | | | | | | |
| CSR | 0.2517 | 1 | | | | | | |
| ln CPA | 0.1329 | 0.1904 | 1 | | | | | |
| Leverage | −0.0515 | 0.1128 | 0.0536 | 1 | | | | |
| Slack | 0.2005 | −0.0226 | 0.0307 | −0.0207 | 1 | | | |
| Advertising | 0.0387 | −0.0266 | −0.0583 | 0.0542 | 0.0542 | 1 | | |
| ln Total Assets | −0.1135 | 0.0852 | 0.3172 | 0.0313 | 0.1314 | 0.0125 | 1 | |
| ln Revenue | 0.0280 | 0.1284 | 0.3725 | 0.0208 | −0.0573 | −0.2020 | 0.6854 | 1 |

### 7.1. CSR–CFP Relationship

Table 3 shows the fixed effects models, static and dynamic OLS models, and system GMM (General Method of Moments). ROA estimates using CSR scores were used for assessing the effect of

supervising dynamic CSR–CFP heterogeneity and relationships. Authentication was performed for the misspecification tests (the second-order serial correlation test (the AR (2) test) and the Hansen test for other identifying restrictions. Authentication made it possible to confirm the suitability of the system GMM model specification. The significant and positive coefficient of the lagged dependent variable validates the persistence of the financial performance, which is significantly dependent on its own past realizations. It is shown in Table 3 that the coefficient of the lagged dependent variable for the system GMM can be found between the pooled OLS and fixed effects. This result can be compared to the work of Blundell, Bond and Windmeijer [118] and Boubakri et al [55], who achieved an unbiased and effective system GMM. This encouraged researchers to utilize and favor the system GMM two-step estimator [119].

**Table 3.** Effects of CSR and Financial Performance (ROA) (N = 134 firms; T = 10; sample period = 2007–2016.

| | Static | | Dynamic | | |
|---|---|---|---|---|---|
| | **Pooled OLS** | **Fixed Effect** | **Pooled OLS** | **Fixed Effect** | **System GMM** |
| **Variables** | **ROA** | **ROA** | **ROA** | **ROA** | **ROA** |
| $ROA_{t-1}$ | | | 0.6140 *** | 0.1490 *** | 0.1980 *** |
| | | | (0.0232) | (0.0293) | (0.0224) |
| CSR | 2.0150 *** | 0.1090 | 0.5910 *** | −0.2980 | 0.2690 |
| | (0.2240) | (0.2430) | (0.1890) | (0.2410) | (0.1800) |
| Leverage | −0.0081 | −0.0041 | −0.0010 | −0.0034 | −0.0009 |
| | (0.0053) | (0.0040) | (0.0041) | (0.0038) | (0.0012) |
| Slack | 0.0771 *** | 0.0196 ** | 0.0311 *** | 0.0104 | 0.0004 |
| | (0.0096) | (0.0081) | (0.0078) | (0.0079) | (0.0065) |
| Advertising | 0.0194 * | −0.1160 *** | 0.0066 | −0.1260 *** | −0.1340 *** |
| | (0.0116) | (0.0327) | (0.0095) | (0.0353) | (0.0379) |
| ln Total Assets | −1.5610 *** | −4.0590 *** | −0.6340 *** | −4.5130 *** | −2.9920 *** |
| | (0.1710) | (0.5450) | (0.1420) | (0.6950) | (0.7390) |
| ln Revenue | 1.1650 *** | 3.8010 *** | 0.3860 ** | 6.8890 *** | 4.9290 *** |
| | (0.1830) | (0.6280) | (0.1510) | (0.8340) | (0.7260) |
| Advertising dummy | 5.0830 ** | 0.1010 | 2.1230 | −1.1700 | −45.9000 |
| | (2.0670) | (2.6480) | (1.5960) | (2.8710) | (32.1000) |
| Constant | −7.9540 | 12.7200 ** | −1.9000 | −11.1700 * | 32.8600 |
| | (2.7720) | (6.1040) | (2.2120) | (6.6900) | (34.6000) |
| Year dummy | Yes | Yes | Yes | Yes | Yes |
| Observation | 1252 | 1252 | 1121 | 1121 | 1121 |
| R-squared | 0.1680 | 0.1040 | 0.4870 | 0.1440 | |
| Number of firms | | 134 | | 134 | 134 |
| AR1 | | | | | −3.300 |
| | | | | | (0.001) |
| AR2 | | | | | 0.230 |
| | | | | | (0.819) |
| Hansen Test | | | | | 40.340 |
| | | | | | (0.411) |
| Difference in Hansen Test | | | | | 6.680 |
| | | | | | (0.351) |
| Number of Instruments | | | | | 51 |

Notes: The standard errors are reported in parentheses, except for Hansen test, AR1, AR2 and Difference-in-Hansen test which *p*-values, ***, ** and * indicate significant at 1%, 5% and 10% levels, respectively. Time dummies are included in the model specification, but the results are not reported to save space. System GMM model is estimated by using the Blundell and Bond (1998) dynamic panel system GMM estimations and Roodman (2009)—Stata xtabond2 command.

To estimate the model of CSR–CFP, we employed dynamic panel data two-step system-GMM estimation. This technique removes the time-invariant unobservable firm-specific effects by determining the first difference for each underlying variable. This process efficiently manages the relationship between the regressors and the residuals. Moreover, with the help of this advanced method, we can eliminate the likelihood of endogeneity by instrumenting differenced equations

through the equations' lagged levels, as well as the variations in the levels, with the lags of the first-differences of the variables.

It was discovered that the following effects were exerted on financial performance by the firm control variables. Both firm size (total assets) and advertising rations exert a negative effect on financial performance. However, leverage ratio has an insignificant impact on ROA for all the models. Nevertheless, organizational slack and revenue produce positive effects on ROA, with an insignificant relationship seen between leverage and ROA. The ambivalence of this effect can be expounded by the presence of higher debt levels that tend not to exact any effects on CFP, even though a high debt level can raise interest costs, a situation that in turn reduces the associated financing cost of corporate strategies [120]. Conversely, debts can similarly produce positive effects, in terms of reducing any issues of agency, by deterring over-investment in slack flows by a self-serving management [66,121,122].

The type of scheme used defines the CSR estimates. From what was observed, certain biases can emerge if dynamic CFP–CSR relationships (fixed-effects) and non-observable heterogeneities (pooled OLS) are dismissed altogether. In the case of positive relationships, these are indicated by the estimates of OLS among CSR and CFP, while neutral associations were observed in accordance with the GMM estimation and fixed-effect findings. Dynamic contexts must therefore be considered when defining CFP–CSR relationships. As biased findings can arise in estimates of OLS and fixed-effect models, we must emphasize the outcomes of the GMM model. In consideration of the associations among CFP and CSR scores, the latter was not found to affect ROA. The given data fail to validate hypothesis H1, implying an absence of direct influences on CFP by prior CSRs. Generally, neutral associations are shown to be present among corporate CSR and CFP activity, and this finding agrees with the results of previous research [16,110,123].

Based on the stakeholder theory, it can be postulated that companies that actively engage in their CSR show a better performance, compared to their competitors, in terms of certain financial performance measures. The stakeholder theory lays the foundation for evaluating the complicated relationships existing between the society and the firm. However, based on the results from the current study, improvements in CSR did not have any immediate financial effect. This result was found to differ with that of Munoz et al. [124], McGuire et al. [36], and Theodoulidis et al. [125]. Additionally, regarding the stakeholder theory, CSR is believed to "aggravate the issues associated with capitalism and ethics" [126] (pp. 413), when added to the responsibilities and financial commitments of a firm. On the other hand, stakeholder theorists recognize a blend of moral and financial consequences in CSR, with a focus on value creation as well as trade via stakeholder relationships, which require trade-offs to address the managerial problems faced by practitioners.

Regarding stakeholder trade-offs, Smith [127] claimed that managers should think carefully about the consequences that result from their CSR activities in order to determine whether those activities violate any rights (imperfect or perfect) [128]. All key stakeholders may potentially benefit from various practical CSR intentions. A few CSR activities may provide benefits to all stakeholders and even avoid violating any rights, but others may need to carry out trade-offs among certain stakeholder groups. Regarding the former, an initiative that would enhance employee compensation could also improve employees' well-being, which is helpful even for the company (as well as its shareholders), thereby enhancing the productivity of the employee. Similarly, one can say that charitable contributions (local charities) could enhance a company's community image, thereby improving sales. One example of such a case, where Smith's philosophy would be relevant, is when the manager considers whether the company needs to donate to a charity or not. Should it be assumed that a donation to the charity will result in minimizing shareholder wealth, a conflict between imperfect and perfect rights arises.

## 7.2. Testing the Moderation Effect of CPA on the CSR–CFP Relationship

Findings regarding the regression scheme used are all displayed in Table 4 for the examination of the proposed hypotheses. As shown, every interaction between CSR and CPA, and how they affect CFP, are presented in Model 3. In evaluating hypotheses 2, Model 3 was utilized, for it represents

a completely quantified framework that delivers a precise schematic of the effect of each variable. Hypothesis 2 can assist in predicting CPA, given that the associations among CSR and CFP can be positively moderated. As CPA increases, positive associations similarly grow stronger among CSR and CFP. Conversely, from our research, it seems that CPA negatively moderates the association, which fails to confirm hypothesis 2. As noted in Model 3, a meaningfully negative coefficient is obtained for interactions between CSR and CPA ($\beta = -1.105$, $p$-value $< 0.05$). This finding implies that CSR influences on ROA grow weaker as CPA increases, which is not consistent with hypothesis 2. Based on the findings from Model 3, we relied on Aiken & West's strategy [129], which plots each meaningful interaction effect to present the corresponding moderating effect. A standard deviation beneath or above the mean would correspondingly represent the lower and higher levels of a moderating variable.

**Table 4.** Effects of CSR and CPA on Financial Performance (ROA) (N = 134 firms; T = 10; sample period = 2007–2016.

| | System GMM | | |
|---|---|---|---|
| **Variables** | **Model 1** | **Model 2** | **Model 3** |
| ROA | 0.2010 *** | 0.1970 *** | 0.2120 *** |
| | (0.0230) | (0.0226) | (0.0261) |
| CSR | | 0.6770 * | 7.7930 ** |
| | | (0.3870) | (3.9300) |
| 1n CPA | −0.1650 | −0.1390 | 7.0970 ** |
| | (0.118) | (0.1210) | (3.4470) |
| CSR*CPA | | | −1.105 ** |
| | | | (0.5270) |
| Leverage | −0.0008 | −0.0008 | −0.0011 |
| | (0.0011) | (0.0012) | (0.0009) |
| Slack | 0.0004 | 0.0005 | −0.0001 |
| | (0.0065) | (0.0065) | (0.0063) |
| Advertising | −0.141 ** | −0.122 ** | −0.148 ** |
| | (0.0379) | (0.0378) | (0.0486) |
| 1n Total asset | −2.9440 *** | −2.8070 *** | −2.3900 *** |
| | (0.7590) | (0.7570) | (0.7190) |
| 1n Revenue | 5.0750 *** | 4.8630 *** | 5.7840 *** |
| | (0.7210) | (0.7900) | (1.0130) |
| Advertising dummy | −48.2600 | −43.6500 | −22.1500 |
| | (35.3900) | (34.8600) | (29.7300) |
| Constant | 36.4300 | 27.3200 | −53.2200 |
| | (37.7800) | (37.0700) | (48.9800) |
| Year dummy | Yes | Yes | Yes |
| Observations | 1121 | 1121 | 1121 |
| Number of firms | 134 | 134 | 134 |
| AR1 | −3.370 | −3.310 | −3.240 |
| | (0.001) | (0.001) | (0.001) |
| AR2 | 0.310 | 0.230 | 0.400 |
| | (0.757) | (0.815) | (0.688) |
| Hansen Test | 40.680 | 40.680 | 34.300 |
| | (0.353) | (0.353) | (0.684) |
| Difference in Hansen Test | 10.040 | 9.000 | 6.360 |
| | (0.123) | (0.173) | (0.384) |
| Number of instruments | 51 | 51 | 51 |

Notes: All models are estimated by using the Blundell and Bond (1998) dynamic panel data system GMM estimations and Roodman (2009)—Stata xtabond2 command. The standard errors are reported in parentheses, except for Hansen test, AR (1), AR (2) and Difference-in-Hansen which are p-values. ***, ** and * indicate significance at 1%, 5% and 10% levels, respectively. Time dummies are included in the model specification, but the results are not reported to save space. The instruments employed in the first-differenced equation are two or more lags of the levels of the endogenous variables, while one lag of the first-difference of the endogenous variables is used as instrument in the difference equation.

Given that significant interactions were identified between CSR and CPA, a further visual assessment [130] was advantageous, as depicted in Figure 2. The recognized CSR effects on ROA for two dissimilar CPA levels (high and low) are shown in Figure 2. For the extent of the involvement of the moderation effects, Figure 2 displays the dominating CSR effect on ROA, when a low CPA level is present, although the associations are weaker, once compared with the higher CPA levels.

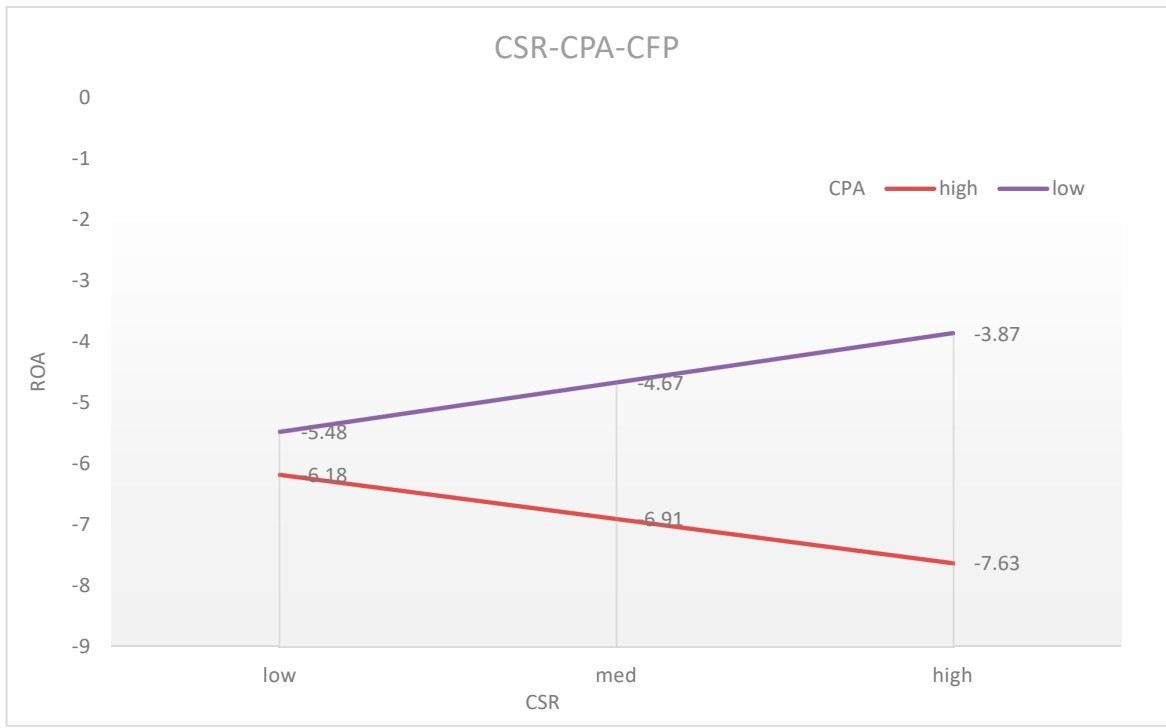

**Figure 2.** Moderation effect of high and low CPA.

When CSR grows from one standard deviation lower than the mean to one standard deviation higher, ROA reduces from –6.18 to –7.63 for those firms that feature a high CPA, whereas ROA grows from –5.46 to –3.87 for firms featuring a low CPA. In other words, when CSR grows from one standard deviation lower than the mean to one standard deviation higher, firms featuring a greater CPA will see ROA reduced by 1.45, whereas ROA increases by 1.59 for firms featuring a low CPA. Figure 2, therefore, shows that as CSR proportions vary from low to high, the associations among ROA and CSR vary from the negative to the positive. This suggests that CPA moderates the relationship between CSR and CFP, particularly given the stronger negative associations observed among CSR and CFP in corporations that feature high CPA levels. The moderating effect of CPA is therefore negative, which is contrary to the hypothesis.

Negative moderation effects, where misalignments exist among corporate CSR and CPA and also where contradictions exist among the company's CSR and CPA policies, can be explained. In accordance with organizational character mechanisms that normally apply to reputational changes, we contest that misaligned activities are seen negatively by various corporate constituencies [62]. When corporate CPA and CSR are misaligned, the situation can be considered as one in which the stakeholders are being intentionally misled, which can also imply a lack of trustworthiness, as perceptions of corporate reputation will eventually be assessed by attentive audiences. In the field of corporate communications, numerous examples can be shown of corporations that advocate proactive CSR initiatives, all the while supporting industrial associations in their lobbying efforts that serve traditional commercial interests. One such example relates to the Global Climate Coalition (GCC), which was established in 1989 by numerous major manufacturers and consumers of fossil fuels, such as auto manufacturers, global oil corporations, and other energy-intensive businesses. The goal of the

GCC was to advance commercial interests in US energy and other climate-related initiatives, as shown in the example of lobbying efforts to convince Congressmen not to impose regulatory measures [131]. Nonetheless, members, such as Shell and BP, began to develop and implement environmental and CSR policies during the 1990s, while funneling investments to alternative and renewable-energy sources. Their publicized positions became difficult to reconcile, which was probably due to the influence of activist campaigns. The coalition was eventually de-established in 2002, as many companies that exited turned out to be more active in industrial coalitions that backed climate-related initiatives, including the World Business Council for Sustainable Development and the US Climate Action Partnership. According to UK activists, in 2009, BP and Shell supported alleged "astroturfing" climate sceptics. American Petroleum Institute members would proffer "upfront resources" as payment for corporate events that hosted protest meetings, in an attempt to contest "legislation that might cap carbon emissions" [14]. This context was revealed after activists began to expose these companies' political lobbying in public. Misaligned activities are intentionally arranged for a variety of strategic motives and can therefore be related to the mutual interaction of CSR and CPA. Conversely, such situations can emerge over time, for instance, when a corporation is unable to mutually coordinate CSR and CPA initiatives with business dynamics in the corresponding field, progressively leading to compromises regarding the initial mutual alignment [14].

## 8. Conclusions

This research provides several positive implications for research, practice, and society, even if its basis is the reported secondary data. Furthermore, it offers useful insights that can be used by diverse stakeholders. This study tried to examine the complex relationship between CSR and CFP, with CPA serving as the moderator. It utilized data from the most admired Fortune corporations between 2007 and 2016. Moreover, it determined no significant associations between CSR and CFP. The relationship between financial performance and corporate social responsibility has been a topic of debate for a significant period [1]. Nevertheless, it appears that the dominant empirical evidence is supportive of the presence of a neutral correlation between the two [16,110,114,123]. However, unlike the predictions of Baron [64] and Orlitzky, et al. [24], the data analysis performed in this study discovered no proof of an existing positive relationship between profitability and strategic CSR measures. Instead, it discovered a neutral association. This variation in the findings can be potentially explained by the indication that a firm can adopt a strategic CSR up to a point where the total costs and the total benefits are equivalent. The results showed that the CPA of a firm can be a proxy for CSR, since it amplifies the influence that CSR has on CFP. Subsequent analysis also revealed that CSR activities provide no real enhancement of the influence of CFP per se (as proven by the non-significant interaction that was seen between the two).

Even though the results of this study revealed an insignificant relationship between CFP and CSR, it is not an indication that CSR engagement is not vital or is unable to deliver any kind of competitive advantage in the long run. Perhaps the significant influence of CSR on CFP is not reflected in the firm's accounting performance but in its non-financial performance. CSR could perhaps function as a motivational force for employees to work harder and enhance their relationship with the stakeholders. The results of this study result suggest that during the initial stages, the cost of CSR activities is more than the benefits acquired by the company in the short term, based on accounting numbers. However, it appears that this investment pays off over time, because employees, governments, customers, and the public need time to appreciate and acknowledge the company's CSR activities.

This study discovered that more socially responsible firms tend to achieve a lower return on their profitability when their lobbying expenditures become higher, in comparison to firms that practice less social responsibility. The findings of this study agree with the contention of Friedman [19], who posited that businesses should not and cannot undertake political responsibilities, since these activities have a negative effect on firm performance. The discovery that CPA has a negative moderating effect on the corporate social–financial performance relationship implies that there is a more complex

association between financial and social performance, compared to what was assumed in previous studies. In order to obtain a more complete understanding of the financial consequences of corporate social performance, there is a need for the examination of how actively a firm takes part in such actions (i.e., the extent of corporate social performance) and the consistency of that performance. Specific recommendations can be suggested to firm managers, who have invested heavily in CSR activities. There is a need for managers to understand CPA's importance. Given the budget limitations for CPA, there is a need for managers to decide whether firms need to engage in CPA a priori for mid and long-term growth. The findings of this study showed that there is a need for companies to conduct minimal lobbying activities, when they have a high social performance and when the firm's CPA strategy is focused on capturing the active political and entrepreneurial opportunities. CPA may not be as effective in influencing public policymaking. Therefore, even if it is managed through CPA, public policy making may only serve as a weak predictor of corporate performance. There might be a need for firms to have access to the government's sustainability policies before they can perform lobbying activities in the governments of other countries. Firms need to be careful when lobbying in the government if there is an expectation that the government will be overthrown sooner or later. Moreover, there is a need for firms to understand the characteristics of the government before deciding which CPA strategy will provide the best results.

*Limitations*

There are some limitations of this study. First, due to the missing data, the size of the unbalanced panel sample was reduced to 134 firms. Therefore, we cannot apply the conclusions of this study to firms that possessed incomplete data during this period. Thus, this study was not able to conduct an exhaustive search. One should note that the actual absence of evidence or the presence of limited evidence of CPA practices does not signify that the industry is not using them. One probable explanation for this is that there is certain missing information, for instance, total amount of lobbying expenditures or donation given to politician. Another possible explanation is that the industry has been applying CPA strategies without announcing it in the public domain, which constitutes a bigger problem. However, there was no difference between this case and the true missing values in this research. Thus, the complete data of firms were used in this study.

Second, it has been indicated by previous studies that the present period's CSR has a stronger correlation to the CFP of the current period compared to the CFP of the later period [24]. It is certain that this concurrent link between CSR and CFP gives rise to concerns related to the ambiguity of the causal relationship found between these two variables. Third, in terms of the measures of CSR engagement, there can still be room for improvement. For instance, due to data limitations, the evaluation of the CSR had to be conducted based on the criteria that Fortune utilized. To a major extent, this database makes it possible to determine the dimensions that are impressive to the internal or external stakeholders, although the design of this particular database was specifically conducted in such a manner. One can provide a more accurate assessment of the CSR engagement strategy of a company by performing a survey at each firm. Fourth, even though there was a need for the firms to specify the issues that they were lobbying for, one cannot reliably connect the currently available lobbying data to specific lobbying issues. In this research, the data on lobbying expenditures also included the other amounts used on lobbying for different purposes. Furthermore, the lobbying variable also failed to capture other kinds of political activities (e.g., PAC activities, unreported personal interactions, campaign contributions, and industry group lobbying). Furthermore, it was difficult to determine the exact time when the results of successful lobbying practices would be observed because of the length and the variability of the policy-making process.

**Author Contributions:** All authors contributed equally to this work. W.L.L. collected and drafted the paper; W.L.L. analyzed the data; J.A.H. and, M.S. reviewed related studies. All authors wrote, reviewed and commented on the manuscript. All authors have read and approved the final manuscript.

**Funding:** This research was funded by Universiti Putra Malaysia grant number GP-IPS 9536600.

**Conflicts of Interest:** The authors declare no conflict of interest.

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
