# Peer review of "Impact of Corporate Political Activity on the Relationship Between Corporate Social Responsibility and Financial Performance: A Dynamic Panel Data Approach"

_sustainability, doi:10.3390/su11010060_

Round 1
Reviewer 1 Report
attached

Author Response
Dear Mr Reviewer,
We sincerely thank the reviewer for constructive criticisms and valuable comments, which were of great help in revising the manuscript. Accordingly, the revised manuscript has been systematically improved with new information and additional interpretations. Please the attachment for your further perusal.

Reviewer 2 Report
GENERAL COMMENTS
The paper is focused on a relevant topic that is interesting enough to be published. In particular, the research investigates the relationship between CSR and financial performance also exploring the mediating role of corporate political activities.
Although authors obtained results contrary to expectations, the research provide useful insights. Hence, I think the paper would be publishable after some minor improvements.
Firstly, i think that introduction is too long and dispersive. You should focus on the research problem better specifying your research objectives and differences that make your paper innovative compared to previous ones. Accordingly, the second comment is on literature review. I think you should introduce the results of previous studies on CSR and financial performance by evidencing how your paper differs from them (e.g. the study of the role of CPA in explaining the difference between CRS and financial performance).
In this way, i suggest you to improve your discussion about results by comparing your results with those obtained in previous studies and better explaining the links with theories (stakeholder and resource dependency).
Finally, i suggest you to split the paragraph on results and discussion by creating a final paragraph devoted only to conclusion in which you discuss the main results obtained, research limitations and implications (as you have already done in discussion and conclusion paragraph).
SPECIFIC COMMENTS
Please avoid the space at the line 578 between the words estimation and This.
Author Response
Dear Reviewer,
We sincerely thank the reviewer for constructive criticisms and valuable comments, which were of great help in revising the manuscript. Accordingly, the revised manuscript has been systematically improved with new information and additional interpretations. Please find the attachment for your further perusal.

Round 2
Reviewer 1 Report
Much improved, though you still need to do a thorough proofreading. Lots of sentences are not clear. For example, the last sentence in the abstract, it should be the high expenditure which worsen the situation, not firms.
Author Response
Dear reviewer,
We regret there were problems with the English. The paper has been carefully revised by professional language editing service to improve the grammar and readability.